# A Dynamic Approach for Early Risk Prediction of Gram-Negative Bloodstream Infection and Systemic Inflammatory Response Syndrome in Febrile Pediatric Hemato-Oncology Patients

**DOI:** 10.3390/children9060833

**Published:** 2022-06-03

**Authors:** José Antonio Villegas Rubio, Pilar Palomo Moraleda, Ana De Lucio Delgado, Gonzalo Solís Sánchez, Belén Prieto García, Corsino Rey Galán

**Affiliations:** 1Pediatric Oncology Unit, Pediatrics Department, Central University Hospital of Asturias, Avda, Roma s/n, 330011 Oviedo, Spain; anade.lucio@sespa.es; 2Pediatric Hematology Unit, Hematology Department, Central University Hospital of Asturias, 330011 Oviedo, Spain; mpilar.palomo@sespa.es; 3Neonatology Unit, Pediatrics Department, Central University Hospital of Asturias, 330011 Oviedo, Spain; solisgonzalo@uniovi.es; 4Primary Care Interventions to Prevent Maternal and Child Chronic Diseases of Perinatal and Developmental Origin Network (RICORS), Instituto de Salud Carlos III, RD21/0012/0020, 28028 Madrid, Spain; crey@uniovi.es; 5Instituto de Investigación Sanitaria del Principado de Asturias, 330011 Oviedo, Spain; 6Clinical Biochemistry, Laboratory of Medicine, Central University Hospital of Asturias, 330011 Oviedo, Spain; prietobelen@uniovi.es; 7Intensive Care Unit, Pediatrics Department, Central University Hospital of Asturias, 330011 Oviedo, Spain

**Keywords:** infection, biomarker, pediatric, oncology, hematology

## Abstract

Background: The aim of this study was to evaluate the usefulness of C-Reactive Protein (CRP), Procalcitonin (PCT), and Interleukine 6 (IL6) biomarkers in predicting the existence of high-risk episodes (HRE) during the first 24 h of fever in pediatric cancer patients. HRE were defined as the presence of Gram-negative bloodstream infections or Systemic Inflammatory Response Syndrome. Methods: The study included 103 consecutive fever episodes in 44 hemato-oncological pediatric patients, from whom samples for biomarkers were taken upon initial evaluation (CRP-1, PCT-1 and IL6-1) and then between 12 and 24 h afterward (CRP-2, PCT-2 and IL6-2). Results: An IL6-1 value higher than 164 pg/mL showed an area under the curve (AUC) of 0.890 (0.791–0.989) and OR of 48.68 (7.92–951.42, *p* < 0.001) to detect HRE in multivariate analysis. A PCT-1 higher than 0.32 ng/mL showed an AUC of 0.805 (0.700–0.910) and OR of 4.55 (0.90–27.84, *p* = 0.076). A PCT-2 higher than 0.94 ng/mL showed an AUC of 0.836 (0.725–0.947) and OR of 13.01 (1.82–149.13, *p* = 0.018), and an increase in CRP between the first and second sample (CRP-2vs1) higher than 291% also showed an AUC of 0.785 (0.655–0.915) and OR of 31.09 (4.87–355.33, *p* = 0.001). Conclusions*:* IL6-1, PCT-2, and CRP-2vs1 showed a strong and independent correlation with HREs in pediatric cancer patients. CRP variations over the first 24 h provide an improvement in predictive models that are especially useful if IL-6 and PCT are not available.

## 1. Introduction

In recent decades, there has been a significant improvement in survival rates among pediatric cancer patients, reaching nearly 85%, or even higher for some kinds of tumors, such as lymphoma or low-risk lymphoblastic leukemia, for which a survival rate of greater than 90% has been achieved [1,2]. Much of this improvement has occurred thanks to an increase in the intensity of treatment, which frequently induces severe immunosuppression, leading to the appearance of infection. According to different reports, the risk of bloodstream infection (BSI) in febrile neutropenic patients is 25–70% [3,4,5], although infections are quite common in non-neutropenic patients as well, with overall BSI rates reported at nearly 15% [6,7].

When signs of infection appear in a cancer patient, especially fever, it would be very useful to have indicators that allow us to discriminate between mild conditions and those that could potentially be severe. Among all the possible infectious conditions, there are two which are especially life-threatening: Gram-negative bloodstream infection (Gr-BSI), and those episodes in which a systemic inflammatory response syndrome (SIRS) develops. Henceforth, we will refer to these as high-risk episodes (HRE).

In recent years, several studies have tried to establish models for predicting infection in pediatric cancer patients with diverse and sometimes contradictory results [5,8,9,10,11,12]. This situation makes it difficult to reach an agreement in stratifying the risk of infection in these patients [13,14]. This lack of consensus leads to variations in management, and thus safety, as well as in the use of resources and patient quality of life among different institutions [12,15,16]. These studies are based on the analysis of clinical and analytical parameters to find indicators that allow us to distinguish the potential severity of a febrile condition. Most of them analyze these parameters at different points during the onset of fever, trying to establish the risk of infection at those moments [5,8,9,10,11,12,17,18,19,20].

Considering the fact that infections, especially the most serious ones, evolve very fast, we hypothesized that variations in the biomarkers C-Reactive Protein (CRP), Procalcitonin (PCT) and Interleukine 6 (IL6) in first few hours of fever could be a good clinical tool in predicting of existence of Gr-BSI and/or development of SIRS (HRE) and could be included when analyzing infectious risk. An approach which takes into account the dynamics of the infectious process could provide more information than an analysis at different static points. To the best of our knowledge, there are no studies using this type of parameters for infectious risk analysis in pediatric hemato-oncology patients. 

## 2. Materials and Methods

We conducted a prospective, observational study in the Pediatric Hemato-Oncolgy Unit of a tertiary university hospital in Oviedo (Spain) from August 2015 to January 2019.

### 2.1. Patients

A total of 103 consecutive episodes of fever in 44 hemato-oncological pediatric patients (aged < 18 years old) were prospectively recorded at the first clinical evaluation for a febrile episode. All of these patients were receiving antineoplastic treatment or inmunosuppresion after allogeneic stem cell transplantation (Allo-SCT). For this study, an individual patient may have had more than one febrile episode.

This study was approved by the Institutional Ethics Committee. Written informed consent was signed by the patients’ parents or guardians and by children over the age of 16.

### 2.2. Methodology

According to our institutional protocol, hemato-oncologic patients with fever should be treated by medical staff within the first four hours of onset. After an initial clinical evaluation, samples for blood count, biomarkers (CRP, PCT and IL6) as well as microbiological samples (central and peripheral blood, urine, and nasopharyngeal tract) were taken. All patients received intravenous empirical broad spectrum antibiotic therapy following our institutional protocol based on recommendations from the Infectious Diseases Society of America [21]. Changes in antibiotic therapy were carried out according to clinical criteria. Subsequent samples were taken after a minimum of 12 h and a maximum of 24 h later to analyze biomarkers, as well as new blood samples for cultures, if fever was still present. 

### 2.3. Definitions:

Fever was defined as a single axillary temperature ≥ 38.3°C for more than one hour or two episodes of fever ≥ 38°C within a 12 h period.Severe neutropenia was defined as an absolute neutrophil count ≤ 500/mm^3^.Gr-BSI was defined as one or more blood cultures positive for a Gram-negative bacterial pathogen either in central or peripheral blood samples.SIRS was defined according to the criteria defined by the International Pediatric Sepsis Consensus Conference (IPSCC) in 2005 [22]. Leukopenia was not considered a criteria for classifying an episode as SIRS if any treatment that could produce it, such as chemotherapy, had been previously received.Henceforth in this article, Gr-BSI and SIRS episodes will be referred to as high-risk episodes (HRE).Samples and their resultant variables obtained in the first evaluation and those of the later 12–24 h evaluation will be labeled with the number 1 and 2, respectively (for example CRP-1 or PCT-2). Resultant variables obtained from the calculation of the percentage of variation between moment 1 and 2 following the formula (Value-2−Value-1)/Value 1 × 100, will be labeled as 2vs1 (example CRP-2vs1).

### 2.4. Statistical Analysis

A descriptive analysis was carried out, providing distributions of relative frequencies and absolute values for qualitative variables and measures of position and dispersion for the quantitative ones. The median of the three biomarkers upon the first evaluation and the 12–24 h evaluation, as well as the median of the variation calculated between these two moments were compared by using non-parametric Wilcoxon test for independent samples, after checking the non-normal distribution of the sample. Optimal cut-off points for these biomarkers were calculated according to the Youden index, which simultaneously maximizes Sensitivity (Se) and Specificity (Sp). Cut-off points, Se, Sp, positive predictive value (PPV), negative predictive value (NPV), area value under the ROC curve (AUC), and the significance of the test were also provided.

Taking into account the optimal cut-off points previously calculated, univariate and then multivariate logistic regression models were constructed for both first evaluation and 12–24 h evaluation, to determine indicators associated with HRE, providing an odds ratio (OR) along with their 95% confidence intervals and significance of the Wald test. Goodness-of-fit was assessed through the likelihood-ratio test, AUC, and Nagelkerke’s coefficient R2. The statistical significance level used was 0.05. Values obtained from the 2vs1 variation were included in the logistic regression model of the evaluation at 12–24 h. Multivariate ROC curve analysis was performed with the biomarkers at first evaluation and 12–24 h evaluation (including in this second one the 2vs1 variables). Finally, taking into account that IL-6 is not available in many institutions, a multivariate regression model was also carried out without taking it into account, in order to provide more adjusted information according to the different available biomarkers at each institution.

Statistical analysis was carried out with the support received from the Statistical Consulting Unit of the Scientific-Technical Services of the University of Oviedo, and R Program (R Development Core Team) version 3.6.3 was used for this purpose [23,24,25].

## 3. Results

### 3.1. Study Population

We analyzed 103 febrile episodes in 44 hemato-oncological pediatric patients (29 females). The demographics are shown in Table 1. The mean age at the time of the febrile event was 7.7 years. Acute Leukemia/Non-Hodgkin’s Lymphoma were the most frequent diagnoses (52.3%). In 62.1% of the episodes, patients were in complete or partial remission. In 50.5% they presented severe neutropenia, and in 34% they were receiving G-CSF at the beginning of the febrile episode.

The clinical outcomes are shown in Table 1. A total of 19 (18.5%) were classified as HRE (11 Gr-BSI, 4 SIRS, and 4 with confluence of both). The most frequent Gram-negative bacteria was *Escherichia coli*, isolated in 6 episodes, followed by *Pantoea aglomerans* in 4 episodes. No infection-related deaths occurred.

### 3.2. Biomarkers

The median and interquartile ranges of biomarkers variables (absolute value of CRP, PCT, and IL6 at evaluations 1 and 2 and percentage of variation 2vs1) were compared between HRE and non-HRE groups (Table 2).

Statistically significant differences between the median of biomarkers in HRE and non-HRE groups were found in CRP-2, CRP-2vs1, PCT-1, PCT-2, PCT-2vs1, IL6-1, and IL6-2.

### 3.3. Estimation of Optimal Cut-Off Point

Table 3 shows the optimal cut-off point to diagnose HRE calculated according to the Youden index, as well as ROC curves at evaluations 1 and 2, and percentage of variation 2vs1. 

The statistically significant cut-off points were 3.5 mg/dL for CRP-2; 0.32 ng/mL for PCT-1; 0.94 ng/mL for PCT-2; 164 pg/mL for IL6-1; and 104 pg/mL for IL6-2. For 2vs1 variations, the optimal cut-off saw an increase of 291.37% and 113.64% for CRP-2vs1 and PCT-2vs1, respectively. No statistically significant cutoff values were found for CRP-1 and IL6-2vs1. For the logistic regression model, 14.4 mg/dL for CRP-1 and 107.3% for IL6-2vs1 were used, as they were the closest ones to statistical significance.

Figure 1, Figure 2 and Figure 3 show the ROC curves for the biomarkers at evaluations 1 and 2, and for the percentage of variation 2vs1, respectively. IL6 at evaluation 1 (IL6-1), with an AUC of 0.89, showed the best accuracy in discriminating HRE, followed by PCT-1 with an AUC of 0.805. PCT-2 (AUC = 0.836) showed the best accuracy at evaluation 2. Regarding 2vs1 values, PCT-2vs1 and CRP-2vs1 showed the best accuracy with an AUC of 0.83 and 0.785, respectively.

### 3.4. Logistic Regression Models

Table 4 shows the univariate and multivariate analysis for HRE diagnosis at first evaluation. An IL6-1 value higher than 164 pg/mL increased the risk of HRE 48.68 times. A PCT-1 value higher than 0.32 ng/mL increased the risk of HRE 4.55 times. CRP-1 was not statistically significant in neither the univariate nor the multivariate analysis. 

Table 4 shows the same analysis if IL6 is not available. PCT-1 value higher than 0.32 ng/mL increased the risk of HRE 8.48 times. CRP-1 was not statistically significant neither in the univariate nor the multivariate analysis. 

Table 5 shows the univariate and then multivariate analysis for HRE diagnosis at the second (12–24 h) evaluation. 

A PCT-2 value higher than 0.94 ng/mL and an increase of CRP-2vs1 greater than 291% are relevant in determining an increased risk of HRE that was 13.01 and 31.09 times higher, respectively. CRP-2, IL6-2, and PCT-2vs1 were statistically significant in the univariate but not in multivariate analysis. IL6-2vs1 was not statistically significant neither in the univariate nor the multivariate analysis.

Table 5 shows the same analysis if IL6 is not available. A PCT-2 value higher than 0.94 ng/mL and an increase in CRP-2vs1 higher than 291% continue being relevant in determining an increased risk of HRE.

A multivariate ROC curve analysis of biomarkers at the first evaluation (AUC 0.904; 95% CI: 0.8–1) (Figure 4) and at 12–24 h evaluation (AUC: 0.915; 95% CI: 0.82–1) (Figure 5) show that both regression models have a very good accuracy for the discrimination of HRE.

## 4. Discussion

The main objective of our study was to find indicators that allow us to make an early prediction of Gr-BSI or SIRS episodes, which we have referred to as high-risk episodes (HRE). These two types of episodes were chosen as the outcome to be predicted due to the fact that they are two of the most serious infectious conditions that a pediatric cancer patient can present during treatment, the latter frequently being the consequence of the former.

Among the possible indicators, analytical biomarkers are of special interest. An ideal biomarker for use in febrile oncologic patients should be able to predict, identify, and thus stratify risk in febrile patients early in their clinical course. In addition, the biomarker should be able to provide robust discrimination of all parameters between mild and serious infections [11].

Several studies have tried to determine the best predictive biomarkers of infection in cancer with contradictory results [5,8,9,10,11,12]. This fact may be due to differences in the cut-off points used, type of episodes to be predicted, and use of only univariate rather than both uni- and multivariate models.

In our study, in addition to the static values of biomarkers, we added their kinetics in the first 24 h in order to try to approximate our predictive model to the changing reality that this type of infectious disease entails. We have not found in the literature any publication that has used these types of variables in pediatric oncology patients.

Among the biomarkers analyzed, the best predictor of HRE at the first evaluation was an IL6-1 value greater than 164 pg/mL. IL6-1 had high sensitivity (92.8%) and specificity (82.5%), with an AUC of 0.89 at this time point.

A PCT-1 value higher than 0.32 ng/mL showed a good correlation with HRE episodes and was very close to statistical significance. This may be due to the fact that IL6-1 behaves as a more powerful predictor that causes an underestimation of the PCT-1 usefulness, since, when performing the multivariate analysis without IL6, PCT-1 has shown to be a more useful predictor and reaches the statistical significance. In fact, for those institutions where IL6 is not available, PCT would be the most useful biomarker. This is supported by its good diagnostic accuracy (AUC = 0.80).

The CRP-1 value did not show statistical significance in predicting HRE. Regarding the biomarkers at the second evaluation (12–24 h), CRP-2vs1 variation was shown as the best predictor of HRE. Neither the CRP-1 nor the CRP-2 values showed statistical significance in predicting HRE. An increase of 291% from CRP-1 to CRP-2 implied a 31.09 times higher risk of HRE. This may be due to the fact that elevated CRP values can occur in these patients due to other types of less serious and slower-evolving infections, as well as non-infectious causes such as mucositis or the oncological disease itself. In these cases, even if the absolute values were high, they would not vary much in a period as short as 12 to 24 h, as can happen in the HRE. In this case, we can observe that the dynamic approach to the infectious phenomenon provides more information than the assessment of static parameters.

PCT-2 was also shown to be a very useful predictor of HRE. In fact, PCT is the only one of the three evaluated biomarkers that showed utility in both the first and second evaluations. This agrees with the good AUC values obtained at both time points. This fact was also shown in a study carried out by Mian et al. They concluded that PCT, and in this paper CRP too, were useful in the risk stratification of febrile neutropenia episodes in pediatric oncology patients [11]. Conversely, this usefulness of PCT seems discordant with the results from a study carried out by Santolaya et al., in which the predictive value of CRP, PCT, and IL8 for severe sepsis was analyzed [10]. The authors concluded that the use of PCT does not provide a significant benefit in the early detection of severe sepsis compared to CRP and IL8. A comparative analysis between studies must be performed cautiously as populations, statistical analyses, and outcomes are different. In addition, our study seems to show that PCT becomes more useful when other biomarkers are not available, as would be the case with IL6. We believe that this information is very useful, since it would allow for the adaptation of infectious risk assessment in each institution, depending on the availability of biomarkers.

IL6-2 and IL6-2vs1 did not prove to be good predictors. This finding differs from those found with IL6-1 and concurs with other studies that show that IL6 is useful as an infection marker at the very initial fever onset, especially in detecting patients with a low risk of presenting an HRE given its high negative predictive value [18,20]. On the other hand, there are others studies where its utility has not been demonstrated [11]. It is worth pointing out that the IL6 value loses its accuracy within 12 to 24 h fever onset due its rapid decrement. This is shown by the fact that median IL6-2vs1 is a negative value.

In clinical practice, according to our results, IL6-1 and PCT-1 values would be the most useful combination of biomarkers at first evaluation, IL6 being the best and earliest predictor of HRE. PCT-2 and CRP-2vs1 would be the most useful ones at 12–24 h evaluation. If we only had one biomarker available in the laboratory, PCT would be the preferred one because it had a good diagnostic accuracy during the first 24 h. But, even if only CRP would be available in an institution, it could provide useful information using its percentage of variation, although with less precocity, since it would require two determinations to get a CRP-2vs1 value.

## 5. Study Limitations

This study presents several limitations that must be taken into consideration: First, it was a single-center study with a small number of subjects and heterogenous diagnosis, all of which could potentially introduce bias. The results of single-center studies are less easily generalized. Second, we have performed an observational study that does not allow any conclusion to be drawn concerning therapeutic interventions. Third the biomarkers levels were analyzed within the first 24 h from the onset of fever, but follow-up measurements were not available. The evolution of the biomarker levels during the first days would have higher accuracy. However, an early severity prediction is more useful in clinical practice in improving patient outcomes.

## 6. Conclusions

Biomarkers with appropriately critical cut-off thresholds may be an important clinical tool for the early prediction of infection. In our study, IL6-1, PCT-2, and CRP-2vs1 showed a strong and independent correlation with HREs in multivariate analysis and, therefore, could be used as reliable predictors of those kinds of severe episodes being useful clinical tools to provide an earlier and more appropriate treatment for these patients. PCT-1 showed a good correlation too and therefore could be used, along with the three above, in the same manner.

We would like to highlight two aspects of the approach of our study: first, the usefulness of biomarker variations over time as a variable that, in the case of CRP, provided an improvement in predictive models; and second, the usefulness of the different information that each biomarker provides at different time points, taking into account their availability.

## Figures and Tables

**Figure 1 children-09-00833-f001:**
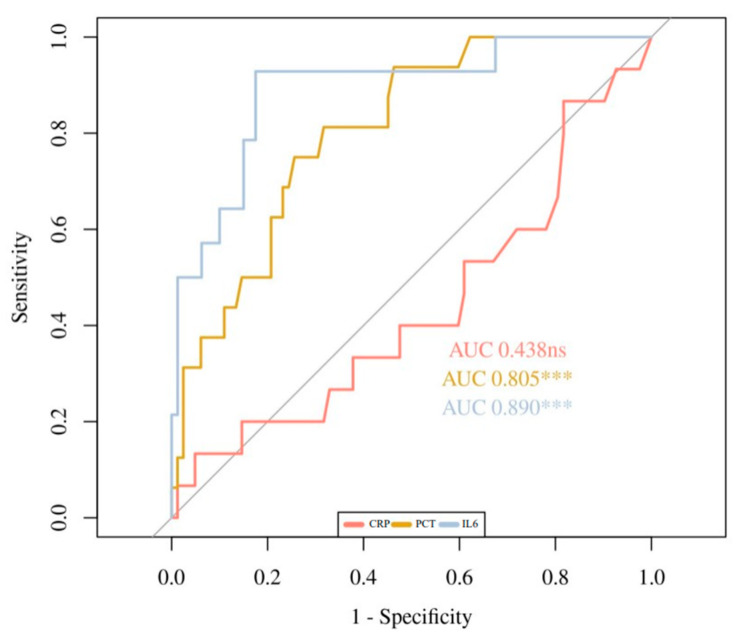
ROC Curves of CRP-1, PCT-1, and IL6-1. ns: non significance, *** *p* < 0.001.

**Figure 2 children-09-00833-f002:**
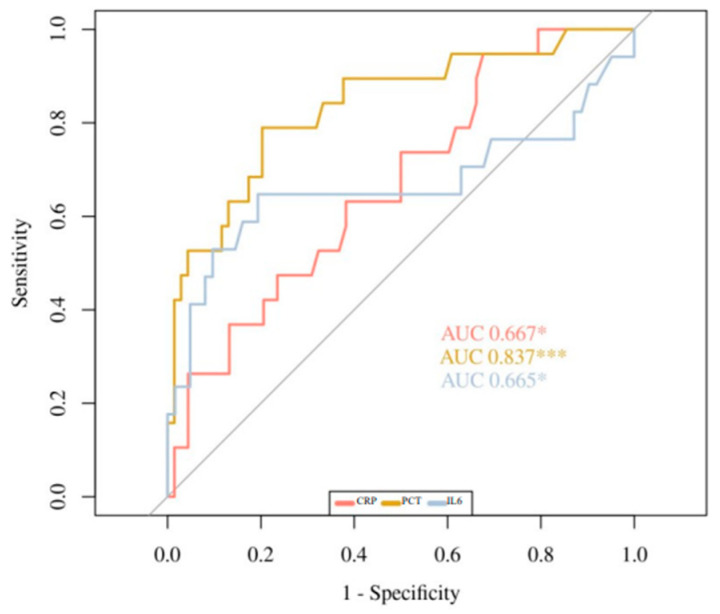
ROC Curves of CRP-2, PCT-2, and IL6-2. * *p* < 0.05, *** *p* < 0.001.

**Figure 3 children-09-00833-f003:**
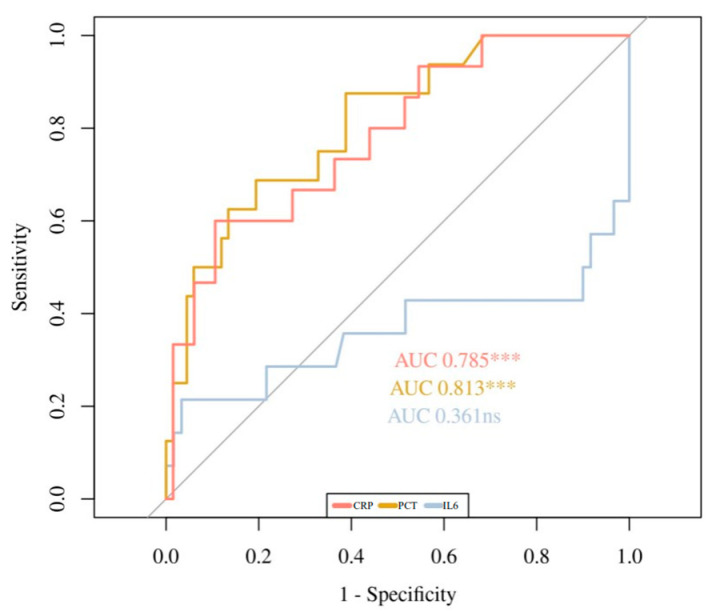
ROC Curves of CRP-2vs1, PCT-2vs1, and IL6-2vs1. ns: non significance. *** *p* < 0.001.

**Figure 4 children-09-00833-f004:**
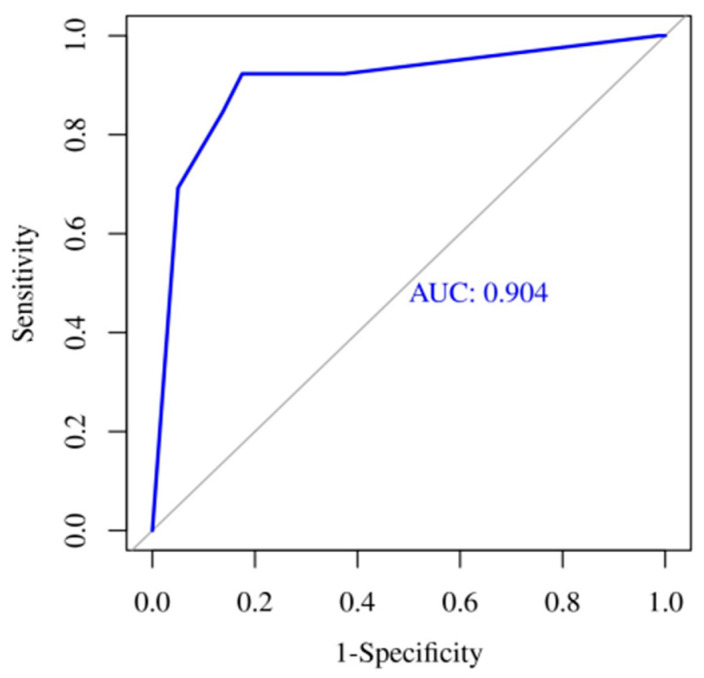
Multivariate ROC Curve for biomarkers at first evaluation.

**Figure 5 children-09-00833-f005:**
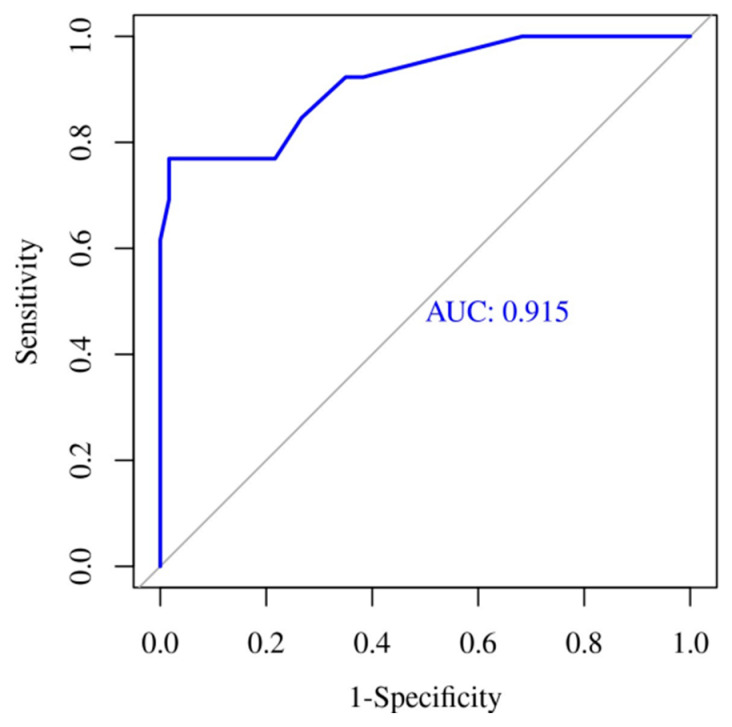
Multivariate ROC Curve for biomarkers at second evaluation (2vs1 variables included).

**Table 1 children-09-00833-t001:** Demographic and clinical characteristics of patients and episodes.

	**Patients** (***n* = 44**)
Female, *n* (%)	29 (66%)
Ethnicity	Caucasian: 44 (100%)
Mean (+/−standard deviation) in years at the first febrile episode	7.7 (+/−5.26)
Solid tumor/HL *n* (%) ALL/AML/NHL *n* (%)	21 (47.7%) 23 (52.3%)
	**Episodes** (***n* = 103**)
Solid tumor/HL *n* (%) ALL/AML/NHL *n* (%)	56 (54.4%) 47 (45.6%)
Disease status: Complete or Partial Remission *n* (%)	64 (62.1%)
Severe Neutropenia *n* (%)	52 (50.5%)
GCS-F: Yes, (%)	35 (34%)
Final diagnoses: HRE *n* (%) Non-HRE *n* (%)	19 (18.5%) 84 (81.5)

ALL: acute lymphoblastic leukemia. AML: acute myeloblastic leukaemia. NHL: non-Hodgkin’s lymphoma. HL: Hodgkin’s lymphoma. GCS-F: granulocyte colony-stimulating factor.

**Table 2 children-09-00833-t002:** Median and interquartile range of biomarkers (absolutes values and variations) in HRE and non-HRE.

	HRE	Non-HRE	*p*-Value
CRP-1 (mg/dL)	2.3 (1.2–5.15)	3.15 (1.52–7.22)	0.451
CRP-2 (mg/dL)	11.1 (5.65–18.75)	6.55 (2.98–12.12)	0.017
CRP-2vs1 (%)	311.1 (77.66–618)	55.52 (8.10–154.7)	0.001
PCT-1 (ng/mL)	0.61 (0.34–3.71)	0.19 (0.13–0.35)	<0.001
PCT-2 (ng/mL)	5.89 (0.96–17.5)	0.28 (0.17–0.76)	<0.001
PCT-2vs1 (%)	290.2 (52.5–1077.3)	16.67 (−6.66–96.67)	<0.001
IL6-1 (pg/mL)	826 (249–4125.75)	76 (35.75–139.25)	<0.001
IL6-2 (pg/mL)	180 (29–341)	44.5 (23.5–86.75)	0.039
IL6-2vs1 (%)	−90.02 (−96.8–3.66)	−47.97 (−71.15–2.18)	0.109

HRE: High risk episode. CRP: C-Reactive Protein. PCT: Procalcitonin. IL6: Interleukine 6.

**Table 3 children-09-00833-t003:** Optimal Cut-off points and Accuracy Indicators for CRP, PCT and IL6 to differentiate HRE from non-HRE.

	CRP-1	CRP-2	CRP-2vs1
AUC (CI 95%)	0.438 (0.267–0.609)	0.667 (0.532–0.802)	0.785 (0.655–0.915)
Se (%)	13.33	94.74	60
Sp (%)	95.12	32.35	89.39
PPV (%)	33.33	28.13	56.25
NPV (%)	85.71	95.65	90.77
Cut-off point	14.4 mg/dL	3.5 mg/dL	291.37%
*p*-value	0.412	0.027	<0.001
	PCT-1	PCT-2	PCT-2vs1
AUC (CI 95%)	0.805 (0.700–0.910)	0.836 (0.725–0.947)	0.812 (0.696–0.928)
Se (%)	81.25	78.95	68.75
Sp (%)	68.29	79.71	80.6
PPV (%)	33.33	51.72	45.83
NPV (%)	94.92	93.22	91.52
Cut-off point	0.32 ng/mL	0.94 ng/mL	113.64%
*p*-value	<0.001	<0.001	<0.001
	IL6-1	IL6-2	IL6-2vs1
AUC (CI 95%)	0.890 (0.791–0.989)	0.665 (0.474–0.855)	0.361 (0.138–0.585)
Se (%)	92.86	64.71	21.43
Sp (%)	82.5	60.65	96,67
PPV (%)	48.15	47.82	60
NPV (%)	98.51	89.29	84.06
Cut-off point	164 pg/mL	104 pg/mL	107.32%
*p*-value	<0.001	0.04	0.11

HRE: High risk episode. CRP: C-Reactive Protein. PCT: Procalcitonin. IL6: Interleukine 6. Se: Sensitivity. Sp: Specificity. PPV: Positive Predictive Value. NPV: Negative Predictive Value. AUC: Area value under the ROC curve.

**Table 4 children-09-00833-t004:** Cut-off values and odds ratio (95% confidence interval and *p*-value) in univariate and multivariate analysis for HRE diagnosis at first evaluation with or without IL-6.

Variable (IL-6 Included)	Cut-Off Point	Univariate OR	Multivariate OR
CRP-1	<14.4 mg/dL	–	–
	>14.4 mg/dL	3.00 (0.39–17.09, *p* = 0.231)	0.18 (0.01–1.81, *p*= 0.18
PCT-1	<0.32 ng/mL	–	–
	>0.32 ng/mL	9.33 (2.73–43.31, *p* = 0.001)	4.55 (0.90–27.84, *p* = 0.076)
IL6-1	<164 pg/mL	–	–
	>164 pg/mL	61.29 (10.90–1159, *p* < 0.001)	48.68 (7.92–951.42, *p* < 0.001)
Variable (IL-6 not Included)	Cut-off point	Univariate OR	Multivariate OR
CRP-1	<14.4 mg/dL	–	–
	>14.4 mg/dL	3.00 (0.39–17.09, *p* = 0.231)	1.10 (0.14–6.67, *p*= 0.92
PCT-1	<0.32 ng/mL	–	–
	>0.32 ng/mL	9.33 (2.73–43.31, *p* = 0.001)	8.48 (2.35–40.57, *p* = 0.002)

CRP: C-Reactive Protein. PCT: Procalcitonin. OR: Odds ratio.

**Table 5 children-09-00833-t005:** Cut-off values and odds ratio (95% confidence interval and *p*-value) in univariate and multivariate analysis for HRE diagnosis at second (12–24 h) evaluation with or without IL-6.

Variable (IL-6 Included)	Cut-off Point	Univariate OR	Multivariate OR
CRP-2	<3.5 mg/dL	–	–
	>3.5 mg/dL	8.61 (1.61–159.79, *p* = 0.042)	1.99 (0.10–76.26, *p*= 0.664
PCT-2	<0.94 ng/mL	–	–
	>0.94 ng/mL	14.73 (4.57–58.36, *p* < 0.001)	13.01 (1.82–149.13, *p* = 0.018)
IL6–2	<104 pg/mL	–	–
	>104 pg/mL	7.64 (2.43–26.35, *p* = 0.001)	4.55 (0.56–50.59, *p* = 0.170)
CRP-2vs1	<291%	–	–
	>291%	12.64 (3.59–49.45, *p* < 0.001)	31.09 (4.87–355.33, *p* = 0.001)
PCT-2vs1	<113%	–	–
	>113%	6.92 (2.19–23.84, *p* = 0.001)	0.53 (0.04–4.29, *p* = 0.578)
IL6-2vs1	<107%		
	>107%	4.83 (0.54–43.68, *p* = 0.133)	0.33 (0.00–18.64, *p* = 0.606)
Variable (IL-6 not Included)	Cut-off point	Univariate OR	Multivariate OR
CRP-2	<3.5 mg/dL	–	–
	>3.5 mg/dL	8.61 (1.61–159.79, *p* = 0.042)	10.69 (0.79–353.54, *p* = 0.113)
PCT-2	<0.94 ng/mL	–	–
	>0.94 ng/mL	14.73 (4.57–58.36, *p* < 0.001)	9.67 (1.81–78.01, *p* = 0.014)
CRP-2vs1	<291%	–	–
	>291%	12.64 (3.59–49.45, *p* < 0.001)	16.81 (3.34–130.48, *p* = 0.002)
PCT-2vs1	<113%	–	–
	>113%	6.92 (2.19–23.84, *p* = 0.001)	1.74 (0.34–9.00, *p* = 0.499)

CRP: C-Reactive Protein. PCT: Procalcitonin. IL6: Interleukine 6. OR: Odds ratio.

## Data Availability

The data that support the findings of this study are available from Central University Hospital of Asturias but restrictions apply to the availability of these data, which were used under license for the current study, and so are not publicly available. Data are however available from the authors upon reasonable request and with the permission of Central University Hospital of Asturias.

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
