# Peer review of "A Dynamic Approach for Early Risk Prediction of Gram-Negative Bloodstream Infection and Systemic Inflammatory Response Syndrome in Febrile Pediatric Hemato-Oncology Patients"

_children, 2022, doi:10.3390/children9060833_

Round 1

Reviewer 1 Report

the article is well-written overall, with few improvement suggestions 

1- i think the last 2 paragraphs in the introduction can be combined together. start with these biomarkers "of biomarkers C-65 Reactive Protein (CRP), Procalcitonin (PCT) and Interleukine 6 (IL6) in ". please mover "

The principal objective of this study is to evaluate the usefulness " at the end of the introduction because this seems not to flow well. 

2- method:  Definitions: it will be easier to follow if you use bullet points. now does not look visually nice and not easy to follow 

3- method: where is the study conducted? you need to include the country as this is very important, especially for cancer. Cancer in China may be different from Europe, Saudi Arabia for example. Also, you are lacking very important information about the race/ethnicity, this is important as well

4- method:  "Statistical analysis was " use indentation for a new paragraph 

6- study limitation and conclusion the font is different from the rest of the manuscript, please fix

Author Response

Thank you

Reviewer 2 Report

In this study, José Antonio Villegas Rubio et al. evaluated the usefulness of C-Reactive 21 Protein (CRP), Procalcitonin (PCT) and Interleukine 6 (IL6) biomarkers in predicting the existence of high-risk episodes (HRE) during the first 24 hours of fever in pediatric cancer patients. They found that IL6-1, PCT-2, and CRP-2vs1 showed a strong and independent correlation with HREs in pediatric cancer patients. CRP variations over the first 24 h provide an improvement in predictive models, especially useful if IL-6 and PCT are not available. Overall, the study was well-designed and presented and has important clinical implications. The following concerns need to be addressed to improve the manuscript.

- Is there any correlation between the underlying hemato-oncological conditions and the three biomarkers tested in the study? It may be informative if the author could stratify patients. 

- Besides the current ROC curves presented, it may be useful to explore multivariate ROC to get better AUC.

- Figure1, CRP misspelled as PCR

Author Response

Thank you
